# Recent Progress in ZnO-Based Nanostructures for Photocatalytic Antimicrobial in Water Treatment: A Review

Ziming Xin [1], Qianqian He [1], Shuangao Wang [1], Xiaoyu Han [1], Zhongtian Fu [1], Xinxin Xu [2] and Xin Zhao [1,*]

[1]  Department of Environmental Engineering, School of Resources and Civil Engineering, Northeastern University, Shenyang 110819, China
[2]  Department of Chemistry, School of Science, Northeastern University, Shenyang 110819, China
*   Correspondence: zhaoxin@mail.neu.edu.cn; Tel.: +86-2483679128

**Abstract:** Advances in nanotechnology have led to the development of antimicrobial technology of nanomaterials. In recent years, photocatalytic antibacterial disinfection methods with ZnO-based nanomaterials have attracted extensive attention in the scientific community. In addition, recently widely and speedily spread viral microorganisms, such as COVID-19 and monkeypox virus, have aroused global concerns. Traditional methods of water purification and disinfection are inhibited due to the increased resistance of bacteria and viruses. Exploring new and effective antimicrobial materials and methods has important practical application value. This review is a comprehensive overview of recent progress in the following: (i) preparation methods of ZnO-based nanomaterials and comparison between methods; (ii) types of nanomaterials for photocatalytic antibacterials in water treatment; (iii) methods for studying the antimicrobial activities and (iv) mechanisms of ZnO-based antibacterials. Subsequently, the use of different doping strategies to enhance the photocatalytic antibacterial properties of ZnO-based nanomaterials is also emphatically discussed. Finally, future research and practical applications of ZnO-based nanomaterials for antibacterial activity are proposed.

**Keywords:** nanomaterial; photocatalyst; antibacterial; zinc oxide; water treatment; antimicrobial

## 1. Introduction

With rapid global population growth, urbanization increasing, illicit misuse of freshwater resources, and continued destruction of the global climate, the increasing demand for clean water is becoming a global concern [1–3]. Seven billion people, more than 15% of the world's people, are facing a shortage of fresh water resources, which even causes them to not have enough fresh water to sustain normal life and productive work [4,5]. Water scarcity is exacerbated by the increasing water pollution from releases of waterborne pathogens, inorganic pollutants, organic pollutants, agricultural chemicals, derivatives of human and animal drugs, and endocrine disruptors [6–8].

Infectious diseases caused by biological contamination such as typhoid fever, dysentery, cholera, and diarrhea are a major cause of death worldwide and continue to replicate at an alarming rate [9]. The extensive use of antibiotics and antibacterial drugs has led to strong drug resistance in viruses and bacteria, which further exacerbates the spread of biological infectious diseases [10]. In addition, the recent epidemics of global security issues such as the COVID-19 virus and monkeypox virus are caused by the spread of viruses that threaten all human beings and have a great impact on human production, life, and health [11–14]. Similar to bacteria and pathogenic microorganisms, epidemic viruses are always difficult to eradicate due to the abuse of antibiotics and various disadvantages of disinfectants [15,16]. In view of the above situation, it is crucial to explore more effective solutions and approaches.

Recent advancements in semiconductor materials and new nanomaterials have blazed new trails for their applications in the fields of photocatalysis and bacterial inactivation [9].

Nanotechnology provides a variety of promising nanomaterials for the field of photocatalytic antimicrobial. Metal oxides have many advantages such as non-toxic, stable, and efficient biological properties, which make them stand out among many nanomaterials and become a research hotspot in this field. Numerous nanomaterials doped with metal oxides such as ZnO [17], $Fe_2O_3$ [18], $TiO_2$ [19], $Ag_2O$ [20], CaO [20], MgO [21], and CuO [22] have been applied as efficient antibacterial agents for both Gram-positive (G+) and Gram-negative (G-) bacteria, such as *Escherichia coli*, *Salmonella enteritidis*, *Streptococcus pyogenes*, *Aeromonas hydrophila*, *Pseudomonas aeruginosa*, *Salmonella typhimurium*, *Fecal intestinal cocci*, etc. [23]. Among the antibacterial agents, ZnO-based nanomaterials are widely recognized as promising antibacterial agents with strong photocatalytic antibacterial activity [24,25]. Nevertheless, the wide bandgap of ZnO is approximately 3.2–3.3 eV, which affects its light absorption ability, resulting in a response only in the ultraviolet band [26]. Previous studies indicated that defects of nanomaterials in photocatalytic antibacterial processes could be effectively improved after being modified [27]. Strategies, such as loading antibacterial agents, loading oxidized nanomaterials, and adjusting the particle size, material microshape, and concentration of ZnO were employed to enhance the antibacterial properties. While exploring the antibacterial ability improvement, the antibacterial mechanism should also be in-depth investigated.

In previous studies, some mechanisms for photocatalytic antimicrobials, such as metal ion release, reactive oxygen species (ROS) generation [28], destruction of cell membranes, internalization of nanoparticles [29], interruption or blockade of transmembrane transport, etc., have been proposed [30,31]. Overall, the purpose of all antibacterial mechanisms is to disrupt the bacterial cell structure and break it down into harmless substances. However, our understanding of the specific process of substance transformation during the photocatalyst-induced antimicrobial process is still very limited, which requires further exploration.

This review is an exhaustive summary of recent research advances on the antimicrobial of ZnO-based nanomaterials. By summarizing previous studies, ZnO-based nanomaterials with excellent antibacterial effects were obtained. From the perspective of material preparation, different preparation methods are reviewed, and both the advantages and disadvantages are compared. Subsequently, the antimicrobial mechanism of ZnO-based nanomaterials is discussed in-depth from both the physical and chemical aspects. In detail, various strategies to enhance the antimicrobial ability of ZnO-based nanomaterials in recent studies are proposed. Finally, temporary deficiencies in the improvement strategy are summed up, and prospects for the future development direction and application potential are presented.

## 2. ZnO-Based Nanostructures Preparation

Among the numerous methods for preparing ZnO-based nanomaterials, the wet-chemical/solution technique has many advantages such as simplicity, rapid operation, and cost savings, which make it a promising method for the preparation of ZnO-based nanomaterials. The advantages and disadvantages of commonly used wet-chemical/solution techniques, such as the sol–gel method, co-precipitation method, microwave-assisted method, and hydrothermal method, are presented in detail in Table 1. The preparatory stage of ZnO nanomaterial growth is fully wetted by wet-chemical/solution techniques, which greatly improves the stability of the materials.

**Table 1.** Advantages and disadvantages of different antibacterial synthesis methods.

| Method | Preparation Shape | Advantages | Disadvantages | References |
|---|---|---|---|---|
| Sol–gel | Nanorods; Nanotubes; Nanobelts; Nano springs; Nano spirals; Nano rings. | (1) Uniform doping; (2) High stability; (3) Low synthesis temperature. | (1) Expensive raw material prices; (2) Longer reaction time; (3) Organics escape. | [32–34] |
| Co-precipitation | Homogeneous and spherical; Nanobelts; Nano springs. | (1) Simple preparation process; (2) Low cost; (3) Short synthesis cycle. | (1) Additional precipitant; (2) High temperature calcination; (3) Uneven dispersion. | [35,36] |
| Microwaves-assisted | Nanorods; Nanotubes; Nanobelts; Nano springs. | (1) High synthesis efficiency; (2) Energy saving; (3) Improved material properties. | (1) Indeterminate form; (2) Large investment; (3) High requirements for equipment. | [37–39] |
| Hydrothermal | Nanobelts; Nano springs; Nano spirals. | (1) Less thermal stress; (2) High particle purity; (3) Controllable crystal shape; (4) Low cost. | (1) Inconvenient to observe; (2) Not intuitive; (3) High equipment requirements; (4) Technical difficulty; (5) Poor safety performance. | [40–42] |

### 2.1. Sol–Gel Method

The sol–gel method is one of the most effective chemical methods for nanocomposites preparation with desired properties and advantages, such as low cost, mild reaction, environmental friendliness, reliability, and simplicity [43]. In the preparation of photocatalytic materials, the materials synthesized by this method have better photocatalytic activity [44]. ZnO nanoparticles were successfully synthesized using the gel-sol method by Hasnidawani [45], and the surface morphology was verified by Fe-SEM images (Figure 1), which was confirmed to have a rod-like structure with a dense particle structure. Varieties of ZnO nanostructures have been discovered, which are in the form of nanorods, nanotubes, nanobelts, nano springs, nano spirals, nano rings, and many more [46]. Among these structures, the rod-like structure is the best nanostructure compared to others due to their one-dimensional nanostructures (such as nanorods, nanowires, and nanotubes) that can facilitate more efficient carrier transport for the decreased grain boundaries, surface defects, disorders, and discontinuous interfaces [47,48]. In the process of preparing ZnO-based nanomaterials by the sol–gel method, the influence of factors such as solution drop acceleration rate, reaction temperature, pH, etc., will have a significant impact on the antibacterial properties of the materials [49]. Effects of various preparation influencing factors on the antibacterial properties of ZnO were tested, and pH was proven to be the most important influencing factor [50]. The reason is suggested to be that the neutral and acidic solution environment is more suitable for $Zn^{2+}$ to function and achieve an antibacterial effect. As long as the optimal pH and preparation temperature are found, the sol–gel method will be one of the effective preparation methods with high efficiency and low cost. Therefore, in recent studies, the sol–gel method is used more in the synthetization of ZnO-based nanoparticles [51–53].

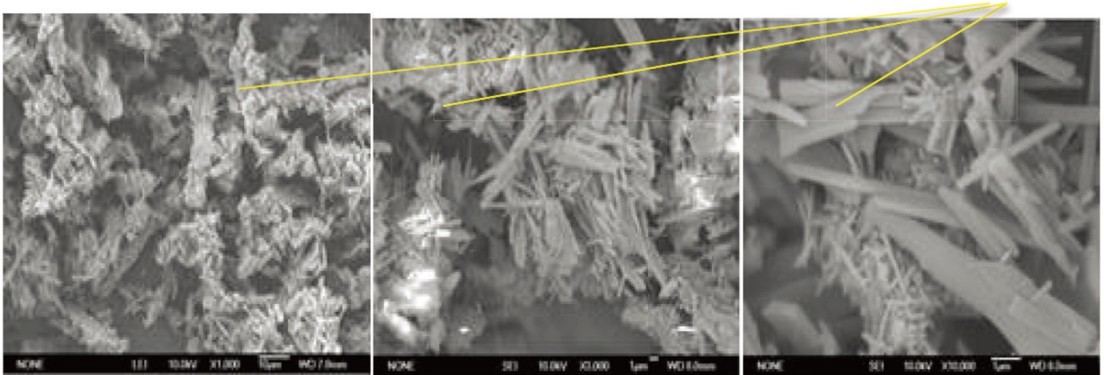

**Figure 1.** FE-SEM micrographs of synthesized ZnO at different magnifications. Reprinted/adapted with permission from Ref. [45]. Copyright © 2022, Elsevier B.V.

### 2.2. Co-Precipitation Method

The co-precipitation method does not require expensive raw materials and complicated equipment, which provides a suitable method for low-cost and large-scale production [54]. Furthermore, in addition to simpler devices, suitable metals, metal oxides, and surfactants are added to change the morphology of the materials [55]. The co-precipitation method was chosen to prepare ZnO-based nanoparticles in an aqueous solution at two different reaction temperatures (50 °C and 70 °C) by Kotresh et al. [56]. The surface morphology of ZnO nanoparticles prepared by the co-precipitation method was observed by scanning electron microscope (SEM) images as shown in Figure 2. It can be seen from the SEM images that the particles are uniformly spherical with a dense and dense structure. The spherical ZnO nanoparticles prepared by the co-precipitation method are favorable for uniform dispersion in the photocatalytic reaction and efficiency improvement of photocatalytic reactions. However, it was found that the droplet acceleration rate had the greatest impact on the antibacterial properties of ZnO-based nanomaterials synthesized by the co-precipitation method [57]. The size of the nanoparticles synthesized by the co-precipitation method is affected by the drop rate of the solution, which will affect the contact area of the nanoparticles during the antibacterial reaction, thereby greatly affecting the antibacterial effect [58]. Therefore, the droplet acceleration rate and particle size need to be carefully considered during synthesis, which is an important part of the success of the co-precipitation method. Due to the advantages of the co-precipitation method with a simple preparation process, the use of this method has gradually increased in nanomaterial synthesis research in recent years [59–61].

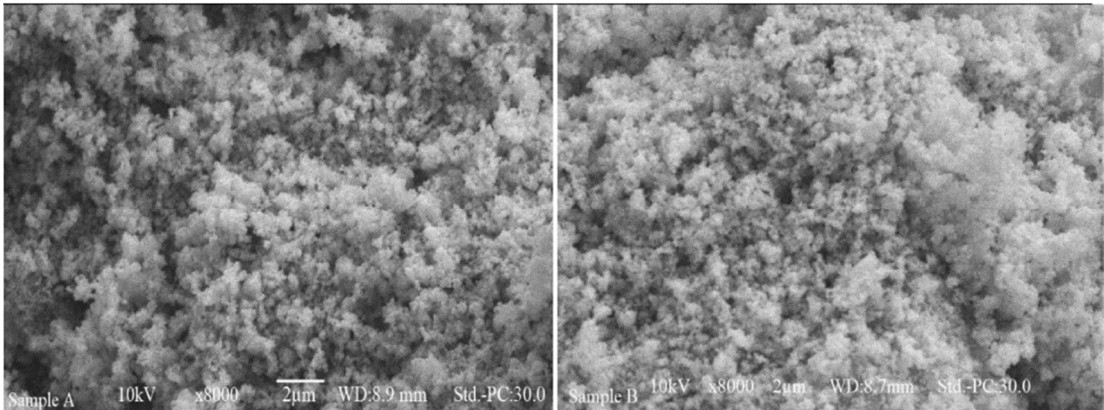

**Figure 2.** SEM images of sample A and sample B. Reprinted/adapted with permission from Ref. [56]. Copyright © 2022, Elsevier GmbH.

### 2.3. Microwave-Assisted Method

The microwave-assisted method is not only an energy-saving, environmentally friendly, and heat-free method, but also with many advantages such as fast synthesis speed and the ability to tune the particle shape [44,62]. ZnO nanoparticles with different morphologies can be synthesized by the microwave-assisted hydrothermal method by adjusting the time and power of microwave irradiation [63,64]. Through microwave-assisted chemistry techniques, ZnO nanostructures with different morphologies were synthesized in different pH reaction mixtures (acidic, basic, or neutral). Furthermore, nanomaterials are synthesized without any heating and addition of surfactants. Hence, obtaining ZnO particles with oxygen vacancies and defects is expected to improve their pollutant degradation behavior due to the fast reaction process and non-stoichiometric synthesis [65]. As shown in Figure 3, the microscopic morphology of the microwave-synthesized ZnO nanostructures was observed by SEM. It can be seen intuitively that the appearance of ZnO nanoparticles changes dramatically with the pH change. In addition, from the XRD analysis in Figure 4, it was demonstrated that the change in intensity and peak width of the two sets of samples (prepared with NaOH and KOH as pH control agents) can be observed as the solution pH changes. The above results showed that the shape of synthesized ZnO nanoparticles is affected by pH changes, which has important implications for the directional synthesis of nanoparticles with diverse morphologies. In addition, several studies have shown that the power of the microwave is the most important factor affecting the synthesis of ZnO-based nanomaterials by microwave-assisted method [66,67]. Moderate-power microwave have been shown to be suitable for the synthesis of ZnO-based nanomaterials with stronger antibacterial capabilities [68,69]. However, the exact power influence mechanism needs to be further explored.

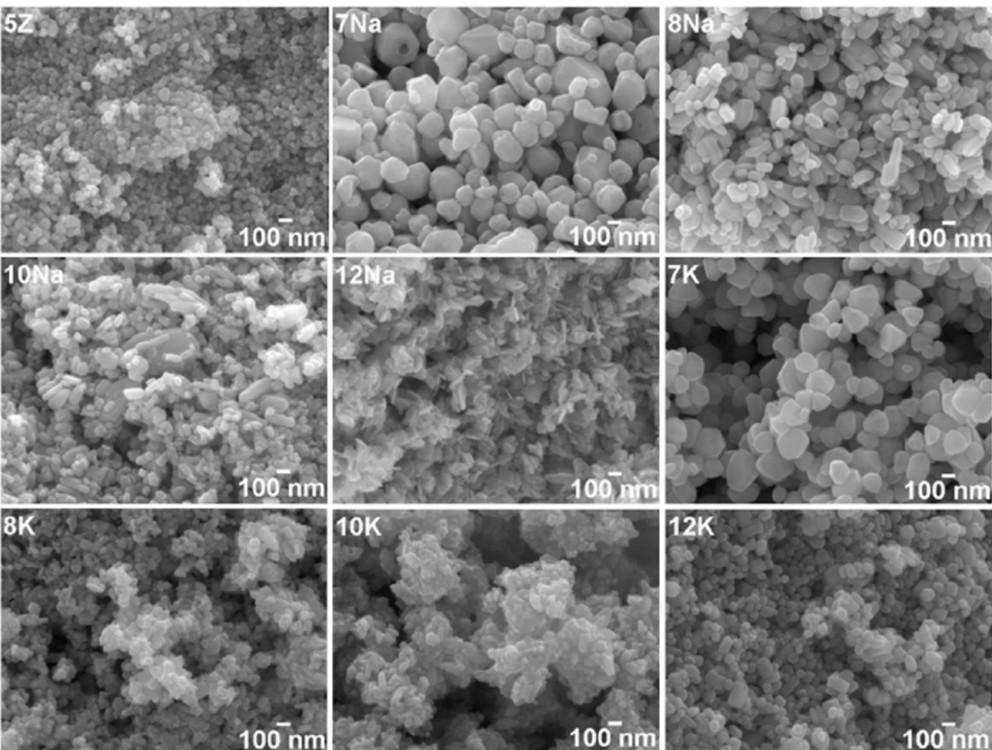

**Figure 3.** Typical SEM images of synthesized ZnO nanostructures at different reaction pH values. Reprinted/adapted with permission from Ref. [65]. Copyright © 2022, Elsevier Ltd and Techna Group S.r.l.

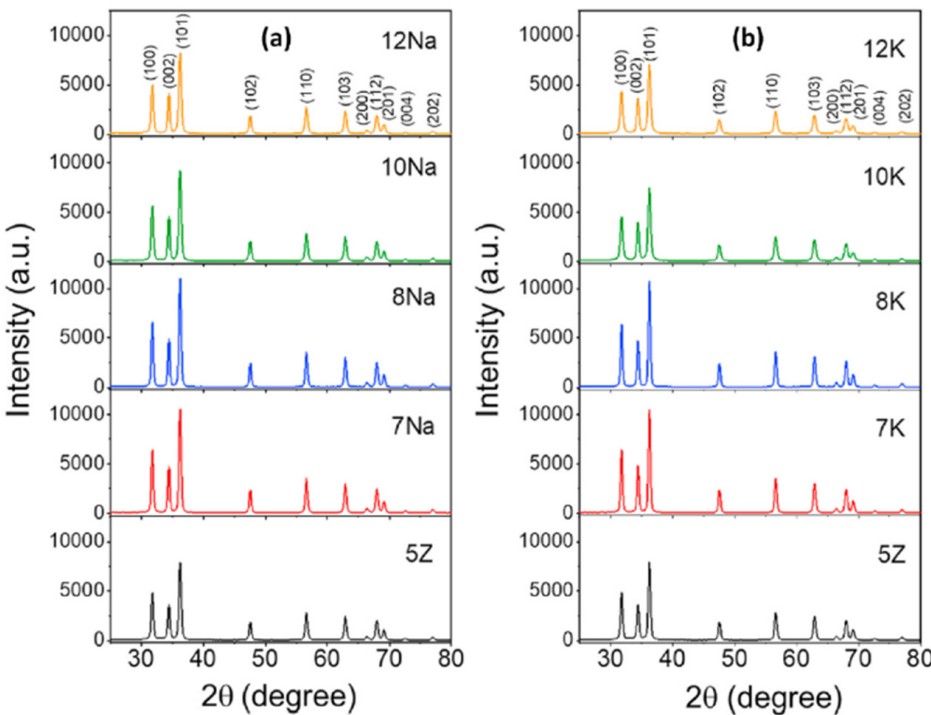

**Figure 4.** Typical XRD patterns of ZnO nanostructures synthesized with different reaction pH ((**a**) NaOH and (**b**) KOH as pH controlling agents, 7, 8, 10, 12 correspond to their respective concentrations). Reprinted/adapted with permission from Ref. [65]. Copyright © 2022, Elsevier Ltd. and Techna Group S.r.l.

### 2.4. Hydrothermal Method

The hydrothermal method is a convenient and highly efficient method, which requires a lower reaction temperature and saves costs [70]. In addition, by adjusting the duration, density, and reaction temperature of the contained substances, the morphology and size of the particles can be controlled [71]. As shown in Figure 5a–c, TEM images of synthesized ZnO particles at precursor concentrations of 5, 10, and 20 mM are revealed. The morphological features of the poorly dispersed nano-ZnO crystals are clearly demonstrated by the TEM images in all cases. At precursor concentrations of 5, 10, and 20 mM, ZnO nanoparticles were observed to have diameters of approximately 4.5, 6, and 8–9 nm, respectively. Furthermore, it was also observed from the images that the size distribution of the nanoparticles was fairly uniform [72]. As shown in the XRD pattern (Figure 6), the crystalline structure of the synthesized ZnO particles after hydrothermal treatment was confirmed. Simultaneously, no impurity peaks were detected from the XRD pattern, indicating that the target substance with a higher purity was successfully synthesized [72]. Furthermore, the preferred orientation of the ZnO particle samples was not seen from XRD patterns, suggesting that ZnO crystals may have the most shapes other than rods or sheets. Based on the above conclusions, the hydrothermal method is a suitable method to prepare ZnO-based nanoparticles with different shapes. In addition, the antibacterial effect of ZnO-based nanomaterials prepared by hydrothermal method is affected by several factors, such as pH, reaction temperature, and dosage ratio [73,74]. The reaction temperature directly affects the structure of the material and changes the antibacterial ability, while the pH changes the surface properties and shape of the material to affect the antibacterial ability [75].

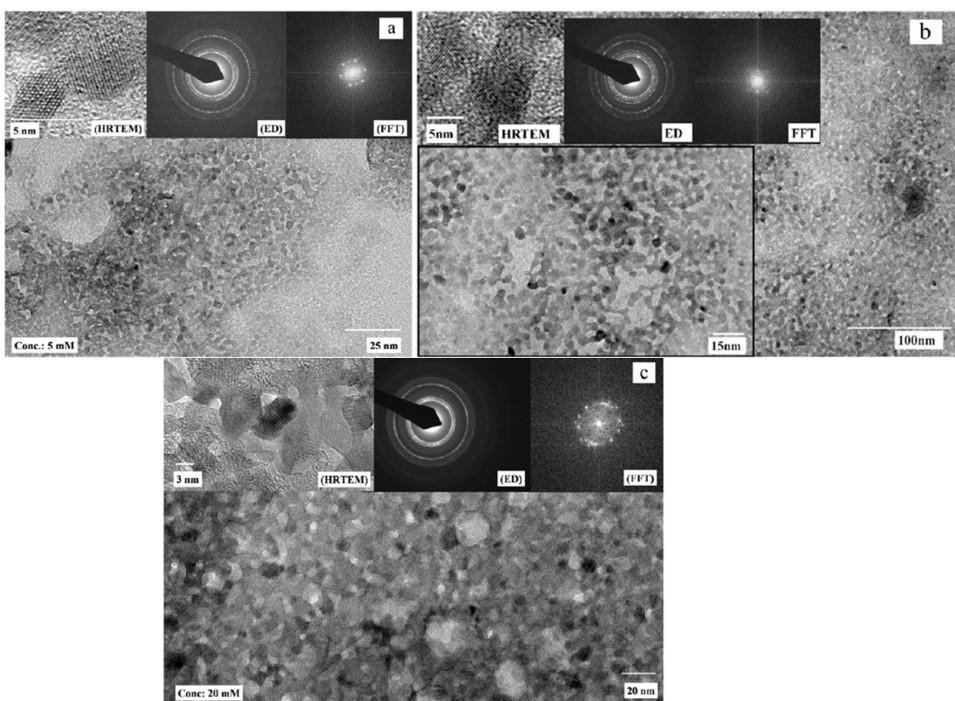

**Figure 5.** Electron microscope image of ZnO nanoparticles: the concentration of precursor is (**a**) 5 mM, (**b**) 10 mM, and (**c**) 20 mM. Reprinted/adapted with permission from Ref. [72]. Copyright © 2022, Elsevier Ltd.

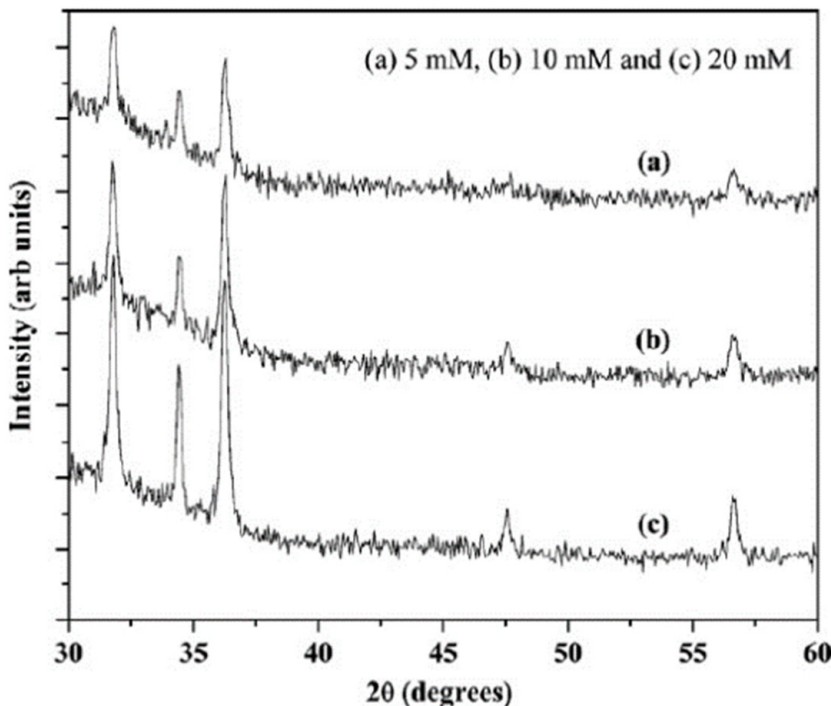

**Figure 6.** XRD patterns of ZnO samples synthesized with different precursor concentrations. Reprinted/adapted with permission from Ref. [72]. Copyright © 2022, Elsevier Ltd.

## 3. Types of Nanomaterials for Photocatalytic Antimicrobials in Water Treatment

A variety of nanomaterials as efficient adsorption materials and catalytic degradation and purification of wastewater will be discussed in this section. As shown in Figure 7, various microbial contaminants and their sources, as well as different nanomaterials for removal applications are revealed [76]. Limitations of single nanomaterials can be overcome

by various strategies such as polymer/metal oxide nanocomposite [77], metal oxide/metal-based nanomaterials [78,79], polymer/metal oxide nanocomposite [80,81], and polymer-structure-based materials [82,83].

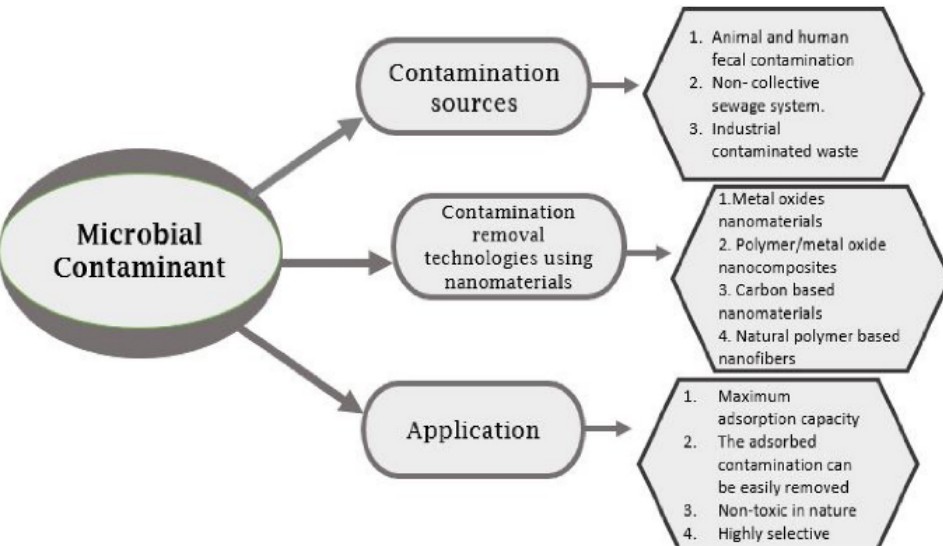

**Figure 7.** Schematic representation of microbial contamination sources and their various nanomaterials for water treatment. Reprinted/adapted with permission from Ref. [76]. Copyright © 2022, Elsevier Ltd.

The nanocomposite is a safe, non-toxic, green and environmentally friendly nanomaterial, usually prepared with ZnO and $TiO_2$ as substrates. The addition of $TiO_2$ and ZnO NPs in some polymers such as polypropylene matrix would increase the dielectric constant of the nanocomposite, thereby enhancing the photocatalytic ability of the material [84]. In addition, the polymer boundary layer transition zone forms a crystalline structure, which increases conductivity and acts as a tuning surfactant [85], which greatly enhances the photoreactivity of the catalytic material. A variety of composite nanomaterials have been reported that can be applied to organic pollutants and microbial contamination removal from water [86]. Moreover, the addition of Ag, zinc, zeolite, and titanium is obtained having better efficiencies in pathogenic pathogens and microorganisms removal from water [87,88]. For example, silver NPs with polyurethane will flexibly remove almost 100% of *B. subtilis* and *E. coli* from water [89,90].

Among the composite nanomaterial antibacterial agents, ZnO-based nanomaterials have the advantages of strong compatibility, green friendliness, lower costs, and simple preparation, so they occupy a large proportion of the related studies in this field in recent years. Studies showed that ZnO has stronger direct interactions with bacterial and pathogen cell surfaces than other semiconductor materials [91]. In addition, ZnO nanomaterials leak $Zn^{2+}$ in solution, which can exacerbate toxicity to bacteria, pathogens, and viruses. Compared with $Cu^+$, $Fe^{2+}$, and $Al^{3+}$, $Zn^{2+}$ showed a better ability to fight microbial contaminants [92]. Therefore, ZnO-based nanomaterials have a better potential to combat microbial contamination, which is of practical significance for the study of antimicrobial contamination. Among types of microbial pollution in the water environment, bacterial pollution is still the most important problem to be solved. Thus, in the next section, the antimicrobial properties and mechanisms of ZnO-based nanomaterials, as well as methods for studying antimicrobial properties are introduced, and the methods for improving their catalytic properties are summarized and discussed.

## 4. ZnO-Based Nanomaterials for Antimicrobial Application in Water Treatment

The antibacterial activity of ZnO-based nanomaterials is greatly affected by the morphology and particle diameter. Considerable methods for the preparation of ZnO-based

nanomaterials with different morphologies have been reported in the literature, such as nanorods [93–96], nano/micro flowers [97–101], microspheres [102], nano powders [103–105], nanotubes [4], quantum dots [106,107], films [107], nanoparticles [108], and capped nanoparticles [109], to understand their application prospects for antibacterial agents. To conduct an in-depth exploration of the antibacterial properties of ZnO-based nanomaterials, the antibacterial research methods in this chapter are firstly introduced, and then the antibacterial mechanism and material improvement strategies are expounded to discuss the latest studies in this field.

### 4.1. Research Methods for Antimicrobial Activities of ZnO Nanostructures

To better study the effect of antibacterial agents, many techniques have been adopted to test antibacterial properties in recent years. As shown in Table 2, various kinds of adsorbents for microbes removal in different water resources are listed, which has great reference significance for antibacterial application studies. In the process of antibacterial ability testing, the accuracy of the results is affected by many factors, such as types of bacterial species and the type of data to be tested, etc. In addition, external factors such as the experimental environment also affect antibacterial detection. Commonly, the temperature of 37 °C and incubation time of 24 h are selected for the tests. Moreover, culture media of Tryptic Soy Broth, Luria–Bertani (LB) broth, Nutrient Agar, and Tryptic Soy Agar (TSA) are commonly used. Meanwhile, depending on the technique used, parameters such as minimum inhibitory concentration (MIC), zone of inhibition (ZOI), colony count, or optical density of bacterial cultures are selected as the basis for the evaluation. The specific application methods and practical application cases of various antibacterial detection technologies are introduced below, which are beneficial for better understanding the antibacterial properties of ZnO-based nanomaterials.

**Table 2.** Various kinds of adsorbents for microbial removal in different water resources.

| Water Type | Sample Type | Microbial Pollution Target | Types of Adsorbents | References |
|---|---|---|---|---|
| sDrinking water | Domestic drinking water | Bacteria; virus; protozoan; <br><br> Microbial contaminants. | Activated carbon; <br> Carbon-graphene; <br> Nano adsorbent; <br> Carbon-graphene. | [110] <br> [111] <br> [112] <br> [113] |
| | Drinking water source | Biological contaminants; <br> Microbial contaminants; <br> *E. coli*. | Nano adsorbent; <br> Iron Oxide Nanoparticles; <br> Mixed matrix. | [114] <br> [115] <br> [116] |
| Wastewater | Lake wastewater | Pathogenic microorganisms. | Zinc oxide; Iron oxide; Silver oxide nanoparticles. | [117,118] |
| | Domestic wastewater | Antibiotics; <br> Pathogenic microorganism; <br> V. fischeri, B. subtilis, E. coli; <br><br> Pathogenic microorganism. | Biochar; <br> Activated carbon; <br> Carbon nanotube; <br> Copper oxide; Zinc oxide; Silver oxide; <br> Titanium oxide. | [119] <br> [120] <br> [121] <br><br> [122,123] |
| | Industrial wastewater | Pathogenic microorganisms; <br><br> Virus. | Graphene oxide nanosheets; <br> Silver nanoparticles; <br> Carbon nanotubes; TiO₂. | [124] <br><br> [125] |
| Laboratory water | Laboratory simulated water | *Enterobacter, Citrobacter, Hafnia*; <br><br> *Klebsiella, Escherichia*; <br><br> *S. aureus, E. coli*; <br><br> Bacteria; virus. | Biochar stabilized; Iron oxide; Copper oxide nanoparticles; <br> $Fe_3O_4$; $SnO_2$; NiO; <br> PAN/boehmite nanofibers. <br> Electro spun; Nanofibers; <br> Silver @ Eggshell; <br> Nanocomposite. | [126] <br><br> [127] <br><br> [128] <br><br> [83,129] |
| | Laboratory pure water | *S. aureus, E. coli, C. albicans*; <br><br> *E. coli*. | Ag nano-embedded pebbles; <br> Nano cellulose; <br> Nanofibers; Granular activated carbon; <br> Graphite flake. | [85] <br><br> [130,131] <br> [132] |

### 4.1.1. Disk-Diffusion Method

The disk-diffusion method is a simple and efficient antimicrobial test, also known as the Kirby–Bauer antibiotic test (KB test) [133]. Mueller–Hinton agar and Brucella blood agar are commonly used as media for this method [133]. During the disk-diffusion method, the

pH of the medium is usually controlled at around 7.2. The bacterial suspension containing a specific concentration was spread on the above agar medium and the experimental environment was required to be absolutely dry [22,24,93,94,97,134–136]. Under sterile conditions, a certain number of ZnO-based nanomaterials were soaked on the filter paper disk using the selected solvent according to the specific requirements of the experiment. Let the disk dry and carefully mount it on the medium in a Petri dish. Solvent-soaked disks were used as controls to ensure the accuracy of the test. Next, the dishes were incubated at 37 °C for 24 h, providing the right conditions for bacterial growth. Subsequently, due to the bactericidal activity of the ZnO-based nanomaterials, no bacterial growth was observed around the disks at specific distances. The minimum concentration at which ZnO-based exhibited antibacterial activity was called the minimum inhibitory concentration (MIC) and the area around the disk with no bacterial growth observed was called the zone of inhibition. The lower the MIC value, the larger diameter of the inhibition zone and the higher antibacterial activity. Based on this, the antibacterial properties of ZnO-based nanomaterials were judged [137,138].

### 4.1.2. Well-Diffusion Method

The medium used in the well-diffusion method is similar to the method described in Section 4.1.1, and the two methods can be used together analogously. In contrast to the disk-diffusion method, the filter paper disk is installed by drilling holes in the media plate. The wells were filled with various concentrations of ZnO nanoparticle suspensions for testing. In addition, the calculation method of MIC and ZOI can also refer to the disk-diffusion method [108,139–142]. Consistent with the disk-diffusion method, sterile conditions are also one of the most necessary environmental conditions for antimicrobial performance testing. Based on MIC and ZOI, the pros and cons of antibacterial properties of ZnO-based nanomaterials can be studied and the efficiency can be evaluated.

### 4.1.3. Antimicrobial Measurements in Liquid Culture Media

During the incubation period, the turbidity of the growth solution increases with bacterial growth. The liquid turbidity and cell proliferation can be measured by periodically measuring the optical density. The technique does not require reagents and special handling [143–147]. The basic principle of this method is to judge the quality of antibacterial performance by observing the absorbance at a specific wavelength in a spectrophotometer and regularly monitoring the corresponding bacterial growth. Umar et al. used ZnO nanomaterials as antibacterial agents to conduct growth inhibition experiments on *E. coli* and achieved excellent experimental results as shown in Figure 8 [101].

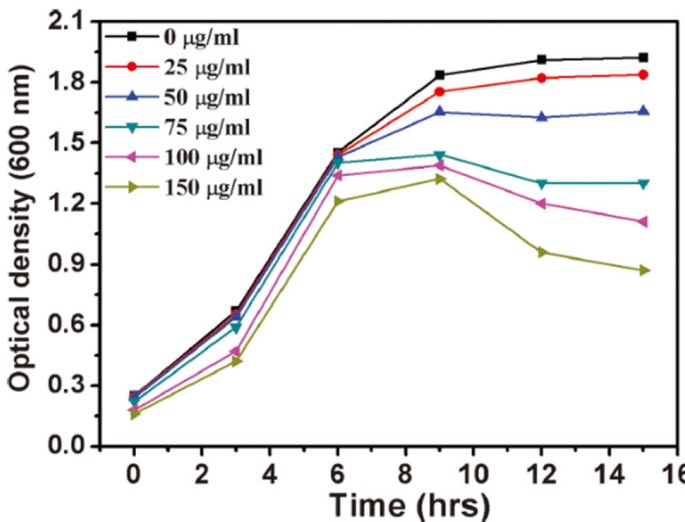

**Figure 8.** Bacterial growth curve of *E. coli* under different ZnO-NFs concentrations.

### 4.1.4. Colony Unit Measurements

Colony unit measurement, also known as the diffusion plate technique, is often used to count the amount of living bacteria cells. ZnO-based nanomaterials are introduced into agar plates or dispersed as suspensions in specific liquid media. The strains were mounted on agar plates and incubated at 37 °C for a specific time. Colony forming units are counted using an appropriate counting method. In addition, the colony forming unit (CFU) value can be used to judge the antibacterial ability of ZnO-based nanomaterials [148–151]. Stankovic et al. [152] calculated the percentage of bacterial cell reduction (R%) using Equation (1).

$$\text{R\%} = \frac{CFU_{control} - CFU_{sample}}{CFU_{sample}} \tag{1}$$

where $CFU_{control}$ = numbers of *CFUs* per milliliter for the negative control, and $CFU_{sample}$ = *CFUs* per milliliter in the presence of ZnO dispersion.

Within a certain range, the above formula can be used to calculate the bacterial concentration, and the antibacterial performance can be investigated based on the calculation results.

### 4.1.5. Microtiter Plate Method

The microtiter plate method, also known as the microplate method, is a method of performing antimicrobial testing by observing changes in a variable number of small test tubes or plates of microwells. Resazurin [153], 2,3,5-triphenyltetrazolium chloride (TTC), crystal violet [4] and 3-(4,5-dimethylthiazol-2-yl)-2,5-diphenyltetra-zolium bromide (MTT) [154], etc., are put into the wells as indicator solutions. Next, known or different concentrations of the test strain was dispersed into the wells of a microtiter plate. A known concentration of ZnO nanomaterials was then dispersed into the wells after dilution in a sterile broth medium. The palate was then incubated at 37 °C for timed intervals. The bacterial cell activity can be judged by the change of absorbance to detect the antibacterial ability.

### *4.2. Mechanisms of ZnO-Based Antimicrobial Nanomaterials*

### 4.2.1. Mechanisms of ZnO-Based Photocatalytic

In photocatalysis, an electron–hole couple is created under light force by reduction or oxidation reactions on the catalyst surface. The photocatalytic degradation mechanism of ZnO-based photocatalyst on pollutants is shown in Figure 9. Photocatalysis occurs when a ZnO-based photocatalyst is illuminated by light with energy greater than its band gap energy [27]. Charge separation is triggered by a light energy absorption process, which excites electrons from VB to CB, leaving holes in VB [155]. Subsequently, the photogenerated e−/h+ supports a move to the ZnO photocatalyst surface. Simultaneously, e− and h+ recombine, which reduces the quantum yield. The level of this recombination rate is affected by many factors, such as the structure of the photocatalyst and the surface modification process of the photocatalysts [156,157]. The ZnO surface is aggregated with reactive e− and h+, which promote oxidation and reduction reactions that generate excess ROS, including superoxide anion ($\cdot O_2^-$) hydroxyl radicals ($\cdot OH$). Furthermore, the redox potential of the CB bottom of ZnO is more negative than that of $O_2/O_2^-$. Therefore, these excited electrons can generate $O_2^- \cdot$. Simultaneously, the top of the VB of ZnO is more positive than the redox potential of $\cdot OH/H_2O$. Consequently, $H_2O$ molecules can be oxidized by these holes to form hydroxyl radicals. These highly reactive radicals ($\cdot OH$, $O_2^- \cdot$) directly oxidize organic pollutant molecules in solutions.

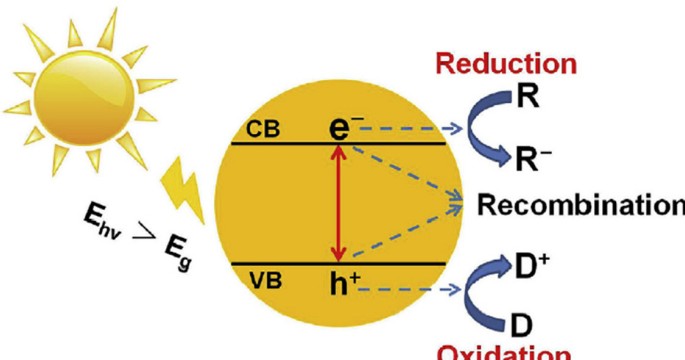

**Figure 9.** Basic mechanism of ZnO photocatalysis. Reprinted/adapted with permission from Ref. [27]. Copyright© 2022, Elsevier B.V.

### 4.2.2. Chemical Effect of ZnO-Based Nanomaterials on Antibacterial

In the process of exploring the antibacterial mechanism of ZnO-based nanomaterials, three main chemical antibacterial mechanisms were obtained: generation of reactive oxygenated species (ROS) [158], release of $Zn^{2+}$ ions [31], and photoinduced production of $H_2O_2$ [159]. The ROS mechanism is basically the same as the photocatalytic mechanism mentioned in Section 4.2.1.

Similar exhaustive studies carried out by Li et al. [31] and Song et al. [160] demonstrated that the toxicity of $Zn^{2+}$ ions to cells is one of the mechanisms for the sterilization of ZnO-based nanomaterials. The concentration of ZnO NPs in ultrapure water in the toxicity test was 5 mg/L. To compare the cytotoxicity, a $Zn^{2+}$ ion solution also prepared in ultrapure water was used (concentration below 0.1 mg/L). TEM images of the treated G− strains of *E. coli* are shown in Figure 10. It is evident that the morphology of *E. coli* was deformed after modification with ZnO-based nanomaterials or $Zn^{2+}$ ion solution. Figure 10b,c shows intracellular fluid leakage due to $Zn^{2+}$ ions and osmotic stress. The experimental results showed that the cytotoxic effects of ZnO NPs and $Zn^{2+}$ ion-treated solutions on *E. coli* were comparable. This fully proves that the release of $Zn^{2+}$ has a positive effect on antibacterial, and it also shows that it is one of the antibacterial mechanisms of ZnO-based nanomaterials (Figure 11).

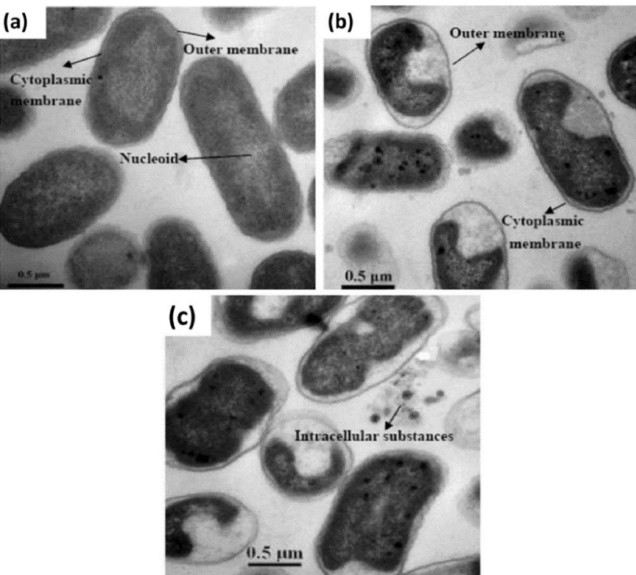

**Figure 10.** TEM images of (**a**) untreated *E. coli* cells, (**b**) treated with ZnO nanoparticles, and (**c**) treated with $Zn^{2+}$ ions solution in ultrapure water. Reprinted/adapted with permission from Ref. [31]. Copyright © 2022, American Chemical Society.

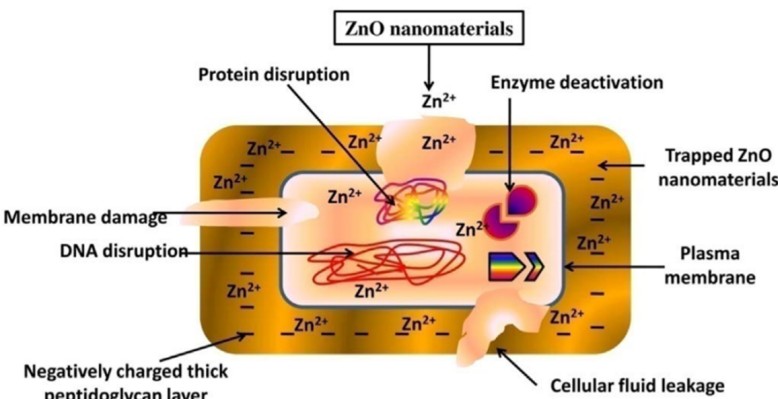

**Figure 11.** Schematic diagram of cell damage to G+ bacteria by $Zn^{2+}$ ions. Reprinted/adapted with permission from Ref. [27]. Copyright© 2022 Elsevier B.V.

Besides the light-induced reactive oxygen species produced by ZnO NPs, many pieces of literature consider $H_2O_2$ as the main substance for antibacterial activity against dermo bacteria [28,161]. Negatively charged ROS cannot penetrate bacterial cell walls, but $H_2O_2$ can also easily penetrate bacterial cell walls. Sawai et al. [162] believed that the $H_2O_2$ produced by ZnO slurry was the main reason for the biocidal mechanism. It can also be hypothesized that after $H_2O_2$ or HO disrupts the membrane, ROS can penetrate the cell wall and enter the intracellular space, thereby enhancing the biocidal effect (Figure 12).

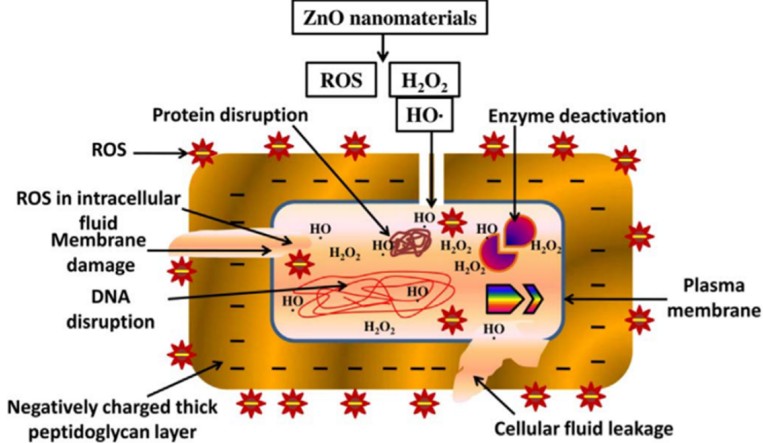

**Figure 12.** Schematic diagram of the damage to bacterial cells caused by ZnO nanomaterials producing $H_2O_2$. Reprinted/adapted with permission from Ref. [27]. Copyright© 2022, Elsevier B.V.

### 4.2.3. Influence of Physical Effects of ZnO-Based Nanomaterials on Antibacterial Performance

In the process of exploring the antibacterial mechanism of ZnO-based nanomaterials, there are three main chemical antibacterial mechanisms: plasma membrane disruption through ZnO interactions [163], cellular internalization of ZnO-based nanoparticles [164], and mechanical damage of the cell envelope [165].

At suitable pH, the bacterial surface is negatively charged due to the dissociation of carboxyl and other functional groups. Meanwhile, ZnO-based nanomaterials are positively charged with a zeta potential of +24 mV [161]. As shown in Figure 13, the opposite charges carried by bacterial cells and ZnO NPs are the reasons for the strong electrostatic attraction between them. Strong electrostatic interactions force particles larger than 10 nm in size to accumulate on the outer surface of the plasma membrane and neutralize the surface potential of the bacterial membrane, resulting in increased surface tension and membrane depolarization. In addition, strong electrostatic interactions can induce bacterial

cell changes such as changes in cell membrane and membrane vesicle structure, rupture, morphological changes, and components, leading to bacterial cell death [166,167]. Since interactions play an important role in the bactericidal effect, surface modifiers and templates of ZnO-based nanomaterials would enhance the interaction with bacterial cell walls.

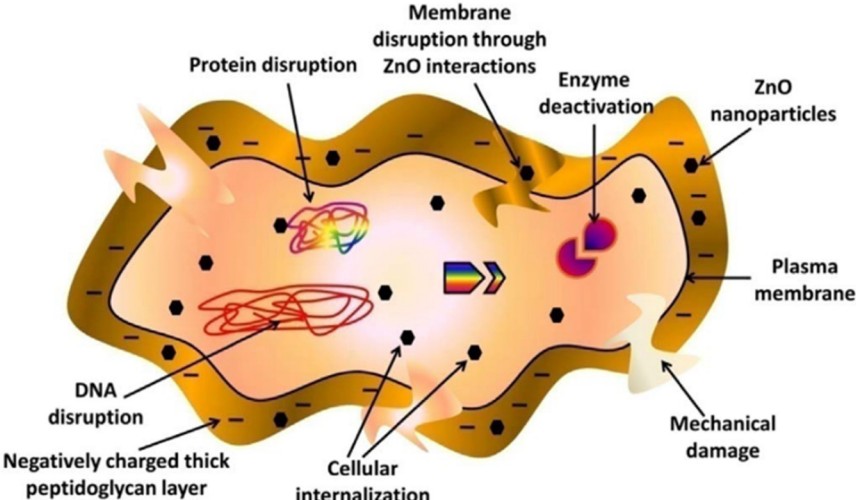

**Figure 13.** Schematic diagram of the physical action of ZnO nanomaterials for sterilization. Reprinted/adapted with permission from Ref. [27]. Copyright© 2022, Elsevier B.V.

Another important mechanism is cellular internalization. Cellular internalization is simply summarized as those nanostructures with a size of less than 10 nm pass through the plasma membrane, accumulate in bacterial cells, and destroy intracellular components such as nucleic acids [164,168–170]. In addition, it has also been suggested that the cellular interaction of ZnO with bacteria can enhance cell permeability (Figure 14) [171]. In conclusion, cellular internalization is one of the physical methods that plays an important role in the antibacterial process of ZnO-based NPs.

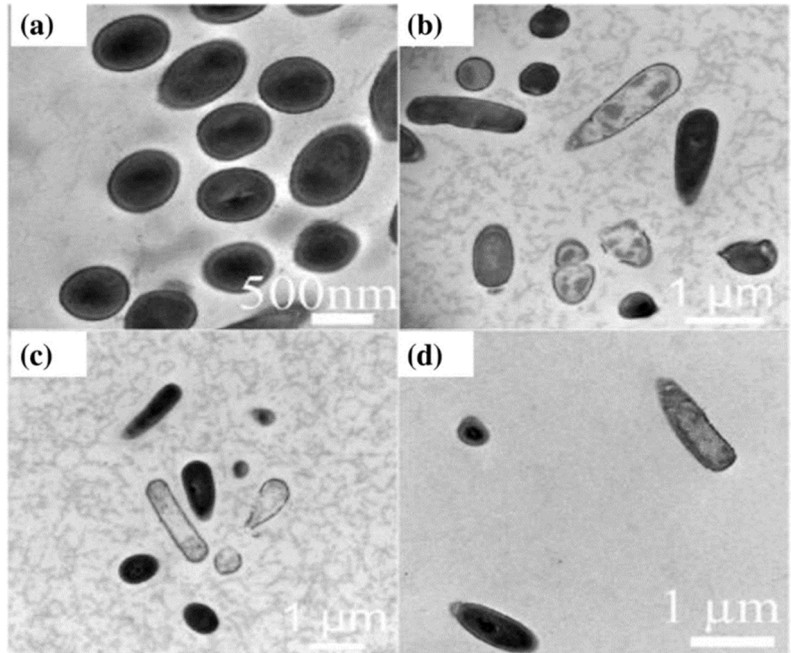

**Figure 14.** TEM images of *B. atrophaeus* (**a**) control, (**b**) ZnO powders, (**c**) ZnO nanorods, and (**d**) ZnO nanoparticles. Reprinted/adapted with permission from Ref. [172]. Copyright © 2022, Elsevier B.V.

The last physical mechanism is to use ZnO NPs to create cell membrane damage to destroy bacterial cells and achieve antibacterial effects. Compared with bulk ZnO materials, the presence of surface defects, uneven surface texture, and rough edges and corners on the surface of ZnO-based nanomaterials can lead to effective abrasiveness, resulting in excessive mechanical damage to bacterial cell membranes [163].

### 4.3. Effects of Radiation Types on the Antibacterial Activity of ZnO-Based Nanomaterials

In the process of photocatalytic antibacterial studies, ultraviolet (UV) light, sunlight, and other visible light are the most common types of radiation. In previous studies, ZnO nanomaterials have always been used for antibacterial testing under UV light due to their high band constraints [173–175]. From the work of Joe et al. [174], an important conclusion was found that the oxygen vacancy of ZnO crystals enhanced the photogeneration of ROS, and ZnO nanoparticles (NPs) with polar facets exhibited the most significant effect of antibacterial activity under UV light stimulation. Furthermore, Ma et al. creatively combined N-halamine-based materials with ZnO to improve the stability of ZnO's antibacterial performance under UV light. As a simple and effective method for nanoparticle modification, this technique can be further extended to the application of ZnO nanoparticles in other polar substrates for antibacterial functionalization [176].

Despite the promising antibacterial performance of ZnO-based nanomaterials under UV-driven radiation, photocatalysis using UV-active semiconductors is difficult due to the limited use of the solar spectrum. Simultaneously, significant progress has been made in photocatalysis using visible-light-active heteronanostructured semiconductors due to their simplicity of use, practicality, reproducibility, reliability, and commercialization [177,178]. Recently, several studies have found that photocatalytic efficiency can be improved by promoting the surface charge transfer reaction of ZnO. In addition, it may also affect the absorption spectrum of many metal oxide nanoparticles including ZnO, which enables the composites to undergo photocatalytic reactions under visible light [179–183]. The antibacterial activity of ZnO-based nanomaterials driven by visible light is of great significance for the development of photocatalytic antibacterials. Compared with ultraviolet light, visible light is more accessible, which makes visible light photocatalysis one of the hottest research fields. In addition, studies have shown that the ability of ZnO-based nanomaterials to excite ROS under visible light will be greatly improved, which also leads to the improvement of antibacterial ability under visible light [184,185]. Therefore, the design and preparation of visible-light-driven ZnO-based nanomaterials will become a meaningful research direction in future explorations.

### 4.4. Strategies for Enhancing ZnO-Based Nanomaterials Antibacterial Activity

#### 4.4.1. Alkaline Earth Metal Doping into ZnO

Common alkaline earth metals including Ca, Mg, Al, and Sr, are frequently used to introduce significantly altered NPs, such as lattice defects and ionic radius differences between metal ions and $Zn^{2+}$ ions, which can improve optical properties and photodegradation activity of catalysts. Taking Sr as an example, it played an important role in enhancing the catalytic degradation ability of commonly used metal oxides such as ZnO, $TiO_2$, which can be used for photocatalytic degradation of organic pollutants in wastewater and photocatalytic antibacterial [186]. Due to the lack of local d orbitals in alkaline earth metals and the presence of local d orbitals in transition metals, the doping of alkaline earth metals is more effective in reducing the optical threshold energy of semiconductors than that of transition metals [187].

Antibacterial activity of Mg-doped ZnO nanostructures was investigated by Okeke et al. [188] towards *E. coli*, *P. aeruginosa* and *Staphylococcus aureus*. The zones of inhibition diameter of the Zn1−xMgxO sample against the selected bacteria pathogen are displayed in Figure 15. The results showed that all samples were susceptible to bacteria. The presence of more reactive sites allows surface defects to create space for ZnO to interact with microorganisms [189]. Therefore, the increase of surface defects in ZnO nanostructures

increases the reaction sites and the rate of interaction with microorganisms. Bacteria are 250 times larger than nanoparticles [105], while the bacterias have a larger relative surface area, which makes it easier for nanoparticles of much smaller size to enter the interior of bacterial cells and cause damage.

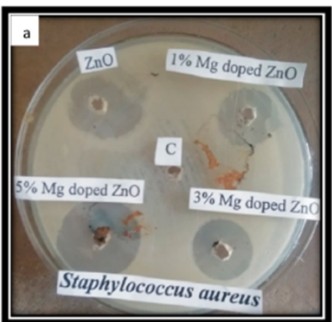
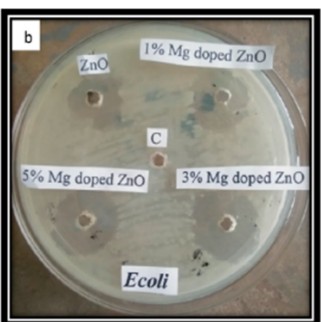
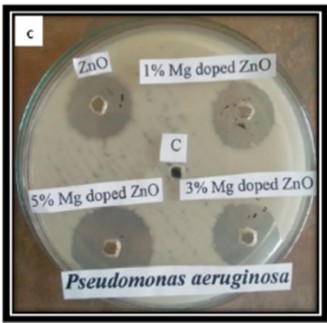

**Figure 15.** (**a–c**) Diameter of inhibition zones of Zn1−xMgxO against bacteria pathogen. Reprinted/adapted with permission from Ref. [188]. Copyright© 2022, Elsevier Ltd.

### 4.4.2. Transition Metals Doping into ZnO

The addition of transition metal ions can generate electronic states in the intermediate bandgap region to change charge separation and recombination kinetics, which is beneficial to enhancing the ability of photocatalytic antibacterial [190]. Numerous common transition metals, such as Co, Cu, Ni, Fe, and Mn, are often doped into ZnO to enhance its photocatalytic antibacterial ability [191].

Fe is the most used metal for enhancing the photocatalytic antibacterial ability of ZnO. Iron has two oxidation states with ionic radii of 0.61 Å and 0.55 Å, which are used as dopants for zinc lattice sites, respectively, since their ionic radii are smaller than those of zinc +2 (0.74 Å) radius. Hence, the doping can be alternative or interstitial to increase the conductivity of the product. Chandramouli et al. [192] doped Fe into ZnO and investigated the antibacterial properties of the composites. As shown in Figure 16, TEM images of Figure 16a pure, Figure 16b Fe doped, and Figure 16c capped ZnO NPs are observed. For undoped and doped ZnO NPs, they are more spherical with dimensions of 17–19 nm. Simultaneously, the agglomeration phenomenon occurs when ZnO is doped with Fe, which is due to the greater surface area and energy. Furthermore, the antibacterial activity against *E. coli* showed that the capped ZnO NPs were less toxic to the organism than ZnO NPs. As shown in Figure 17, the XRD patterns associated with various different ZnO-based nanomaterials are displayed. Iron doping of ZnO reduces the grain size, resulting in a further increase in the grain size of glucose-terminated ZnO nanoparticles, which is consistent with previous reports [193]. Therefore, Fe-ZnO nanoparticles have good antibacterial activity.

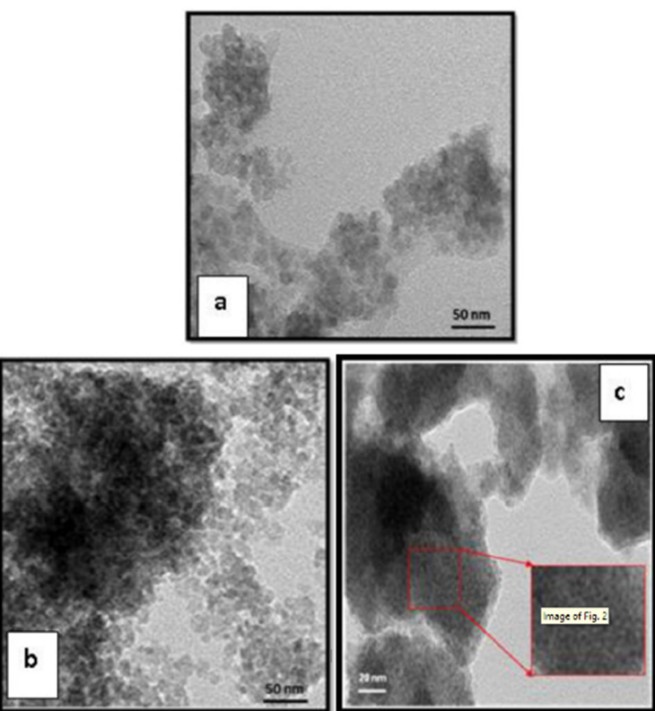

**Figure 16.** TEM images of (**a**) pure, (**b**) Fe doped, and (**c**) capped ZnO nanoparticles. Reprinted/adapted with permission from Ref. [192]. Copyright© 2022, Elsevier B.V.

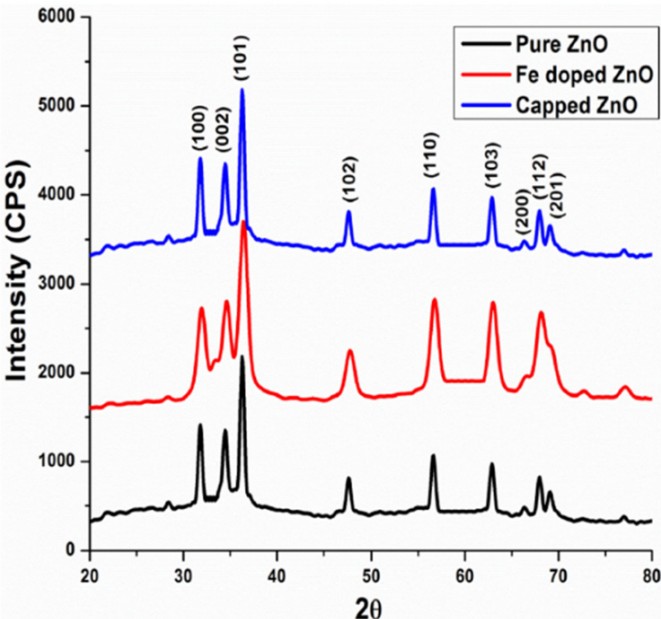

**Figure 17.** XRD patterns of various ZnO-based nanomaterials. Reprinted/adapted with permission from Ref. [192]. Copyright© 2022, Elsevier B.V.

### 4.4.3. Noble Metals Doping into ZnO

Due to the formation of Schottky barriers at the metal–semiconductor interface, noble metal ions (such as Au, Ag, Pd, etc.) doped on the ZnO surface are considered to be excellent photogenerated electron traps. In addition, noble metals delay electron–hole recombination by preventing photoexcited electrons from returning to the ZnO surface, which greatly enhances the photocatalytic antibacterial ability of the composites [194]. Of all the precious metals, silver is the most stable and suitable dopant with good thermal conductivity and

electrical conductivity, which will better play the photocatalytic effect of the composite material. Therefore, it has potential as a catalyst. The surface plasmon resonance (SPR) properties of silver also contribute to visible light absorption and subsequent electron–hole pair generation for the degradation of contaminants in water and for antibacterial [195].

Ye et al. [184] reported the synthesis of a series of ZnO/Ag$_2$MoO$_4$/Ag(ZAA) samples with theoretical molar ratios of ZnO and Ag$_2$MoO$_4$ of 20:1, 30:1, and 60:1 by ultrasonic-assisted hydrothermal synthesis, and named the corresponding products as ZAA-20, ZAA-30, and ZAA-60 to investigate the optimal Ag$_2$MoO$_4$/Ag loadings. As shown in Figure 18, the characteristic diffraction peaks of ZnO and Ag2Mo4 can be clearly found in the XRD patterns, which indicates that the ZnO/Ag$_2$Mo$_4$ composite was successfully synthesized. The sharp diffraction peaks reveal the ultra-high crystallinity of the ZnO-based nanocomposites. In addition, with the increase of the molar ratio of ZnO to Ag$_2$MoO$_4$, the diffraction peak intensity of ZnO on the (002) and (110) crystal planes gradually weakened, while the diffraction peak intensity of Ag$_2$MoO$_4$ gradually increased. The antibacterial properties of ZnO nanosheets and ZAA nanocomposites against different contents of G− *E. coli* and G+ *S. aureus* were evaluated by the visible light electroplating counting method. In Figure 19, the bacterial cell numbers of all nanocomposites were shown to decrease with increasing contact time, and the photocatalytic antibacterial activities of the four ternary ZAA nanocomposites were much better than that of pure ZnO sheets. The experimental results demonstrate that the addition of noble metal Ag will significantly improve the antibacterial properties of ZnO nanomaterials.

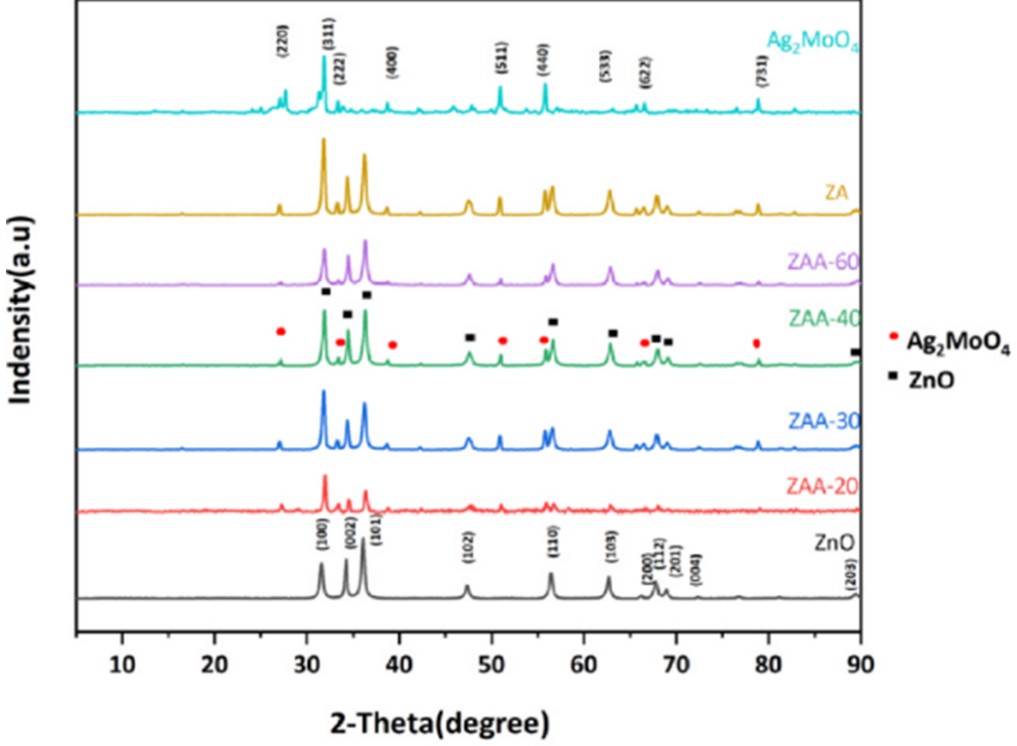

**Figure 18.** XRD patterns of ZnO, Ag$_2$MoO$_4$/Ag, ZA, and ZAA nanocomposite with different proportions. Reprinted/adapted with permission from Ref. [184]. Copyright© 2022, Elsevier B.V.

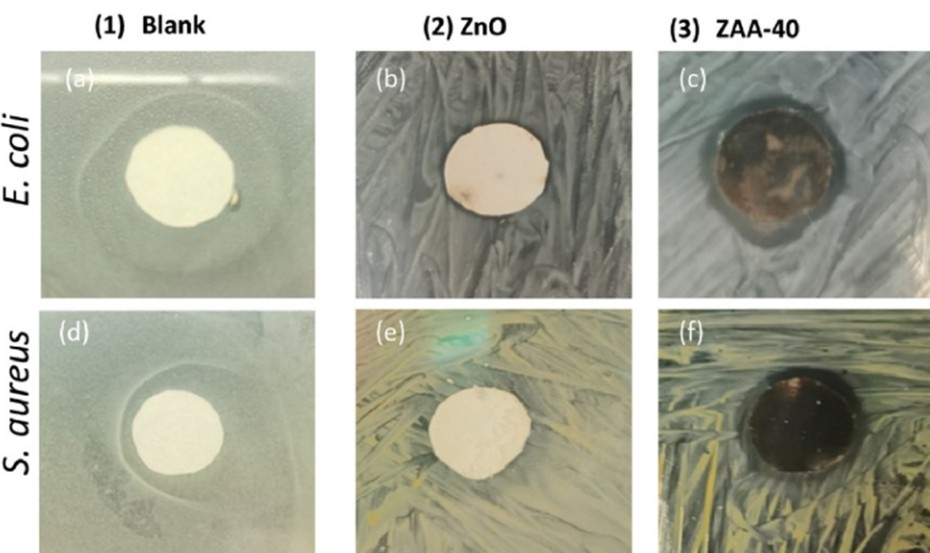

**Figure 19.** Photographs of antibacterial test results of ZnO, Ag2MoO4/Ag, and ZAA samples against Escherichia coli (**a–c**) and Staphylococcus aureus (**d–f**). Reprinted/adapted with permission from Ref. [184]. Copyright© 2022, Elsevier B.V.

### 4.4.4. Rare Earth Metal Doping into ZnO

Doping rare earth metals into ZnO can improve the ability of the composite in trapping photogenerated carriers and reducing electron–hole recombination, which can enhance the photocatalytic antibacterial ability. In the rare earth doping process, f-orbital doping is the most common and efficient way, which can improve the photocatalytic activity by enhancing the adsorption of pollutants on the catalyst surface, while reducing the band gap energy to the visible light range [196]. Lanthanide ion doping is considered a versatile strategy to tune the optical response and improve the photocatalytic performance of ZnO. Lanthanides are composed of 17 elements in the periodic table, including Sc, Y, La, Ce, Pr, Nd, Pm, Sm, Eu, Gd, Tb, Dy, Ho, Er, Tm, Yb, and Lu. Lanthanides have attracted much attention due to their multifunctional properties resulting from their unique f-orbital structures, and due to the f-f or f-d intra-electron transitions, lanthanides are considered candidates for luminescent centers in doped materials, which is beneficial to prolonging the effective response time [197–199].

Doping ZnO with $Ln^{3+}$ and $Ce^{4+}$ ions can convert the magnetism from diamagnetism to ferromagnetism, improve the n-type conductivity, enhance the photo response, increase the concentration of free electrons in the CB, and increase the electron mobility [200–202]. A novel Z-type $ZnO–CeO_2-Yb_2O_3$ heterojunction photocatalyst was prepared for the first time by Tauseef et al. [203], and its physical, photocatalytic, and antibacterial properties were investigated. Growth samples were tested for antimicrobial properties against *E. coli* and *S. aureus.* The effects of operating parameters such as catalyst dosage, dye concentration, and solution pH on the photocatalytic performance of the nanocomposites were investigated. The ZOIs of *S. aureus* and *E. coli* along with the standard antibiotic ciprofloxacin are shown in Figure 20a,b. The synthesized nanocomposites exhibited good activity against both bacteria with a ZOI > 6 mm, but higher activity against *E. coli* with a ZOI of 14 mm shown in Figure 21a,b. Positively charged heavy metal ions such as $Zn^{2+}$, $Ce^{4+}$, and $Yb^{3+}$ can be released from the surface of the nanocomposite to interact with negatively charged microbial cell membranes. The entry of these metal ions into the cell membrane reduces the capacity and permeability of proteins, which in turn leads to the death of microorganisms such as bacteria and viruses. The above antibacterial action mechanism can be visualized in Figure 22C. In conclusion, the nanocomposites doped with rare earth ions are effective materials for preventing diseases caused by *S. aureus* and *E. coli.*

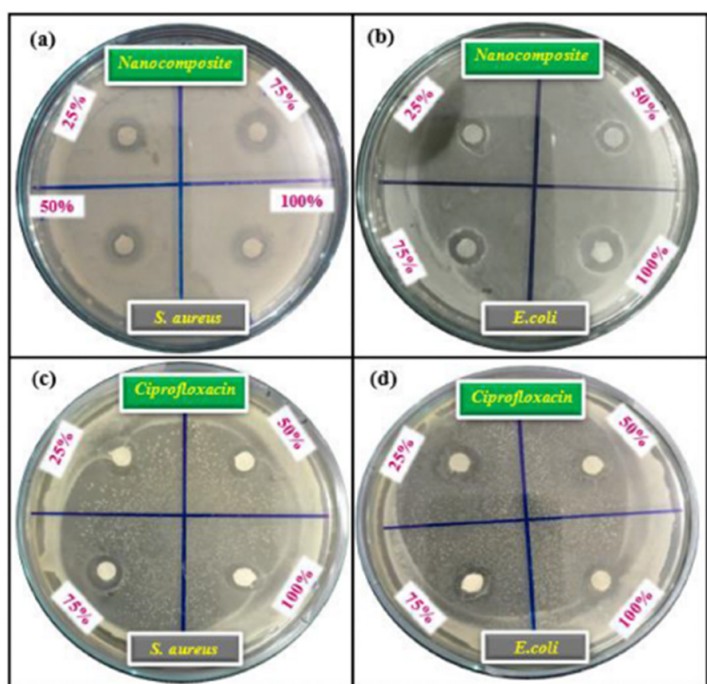

**Figure 20.** Antibacterial effect of ZnO-CeO$_2$-Yb$_2$O$_3$ nanocomposites: (**a**) gram-positive *Staphylococcus aureus* bacteria, (**b**) gram-negative *Escherichia coli* bacteria at different concentrations, and (**c**,**d**) Standard antibiotics Ciprofloxacin against *S. aureus* and *E. coli*. Reprinted/adapted with permission from Ref. [203]. Copyright© 2022, Elsevier Masson SAS.

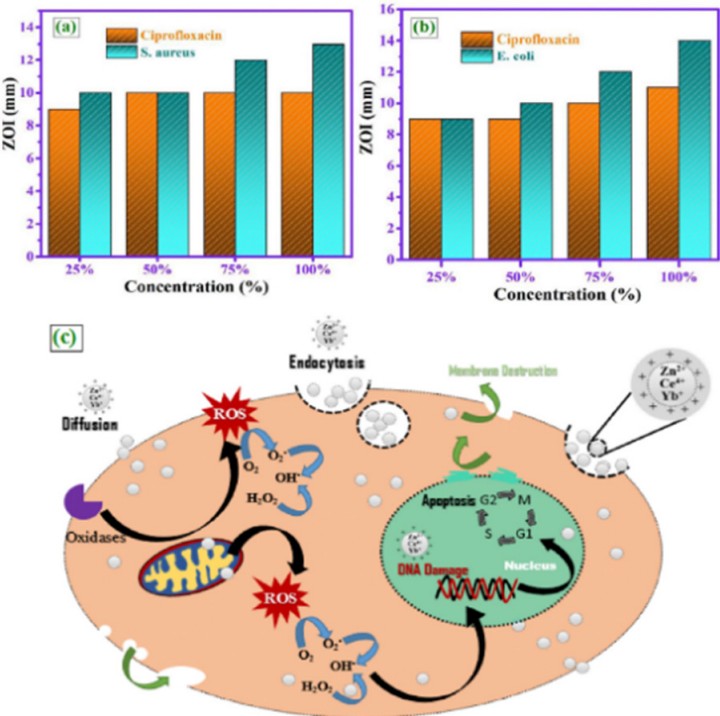

**Figure 21.** Comparison of the zone of inhibition (ZOI) against different species of bacteria (**a**,**b**), mechanism of antibacterial activity of ZnO–CeO$_2$-Yb$_2$O$_3$ nanocomposite (**c**). Reprinted/adapted with permission from Ref. [203]. Copyright© 2022, Elsevier Masson SAS.

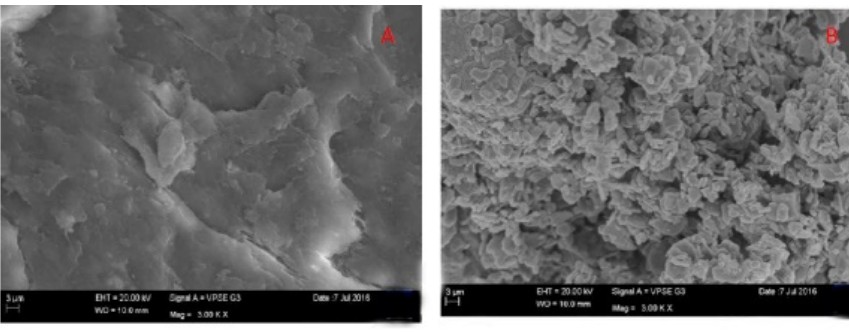

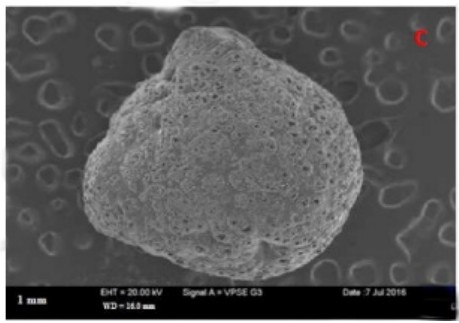

**Figure 22.** SEM images of various prepared CS/PVA/ZnO-related materials and precursors: (**A**) pure chitosan, (**B**) ZnO, and (**C**) CS/PVA/ZnO. Reprinted/adapted with permission from Ref. [204]. Copyright© 2022, Elsevier B.V.

### 4.4.5. Organic Antimicrobial Agents Doping into ZnO

Studies have shown that the composites obtained by co-doping and fusion of organic antibacterial agents and ZnO nanoparticles exhibited stronger antibacterial activity than ZnO nanoparticles alone [205,206]. The organic antimicrobial agents are usually immobilized or embedded on the ZnO surface. Taking chitosan (CS) as an example, it is an abundant natural biopolymer derived from the deacetylation of chitin in crustacean shells and can be made into films, fibers, beads, and powders. Cationic polymers are generally antimicrobial [207,208]. In general, antibacterial activity depends on molecular weight (Mw), degree of deacetylation, temperature, and solution pH [209,210].

Gutha et al. [204] used CS and ZnO as raw materials to prepare a new composite material chitosan/poly(vinyl alcohol)/zinc oxide (CS/PVA/ZnO), which was used as a novel antibacterial agent with wound healing properties. CS/PVA/ZnO was proved to be an effective antibacterial nanomaterial after being analyzed by various characterization methods. SEM images of various prepared CS/PVA/ZnO-related materials and precursors are shown in Figure 22A–C. The surface of sole chitosan was obtained to be smooth. The sole ZnO nanoparticles showed nanosheet-like morphology. The surface of sole CS/PVA/ZnO microbeads presents a certain pore structure, and the surface of the microbeads is rough, which is conducive to exerting the ability of photocatalytic antibacterial. The antibacterial activities of CS, CS/PVA, and CS/PVA/ZnO are shown in Figure 23. The diameter of the inhibition zone against *E. coli* cultures (G-) was 10 mm in the CS group, 14 mm in the CS/PVA group, and 19 mm in the CS/PVA/ZnO group (Figure 23A). Likewise, *S. aureus* cultures (G+) had a diameter of 12 mm in the CS group, 15 mm in the CS/PVA group, and 20 mm in the CS/PVA/ZnO group (Figure 23A).

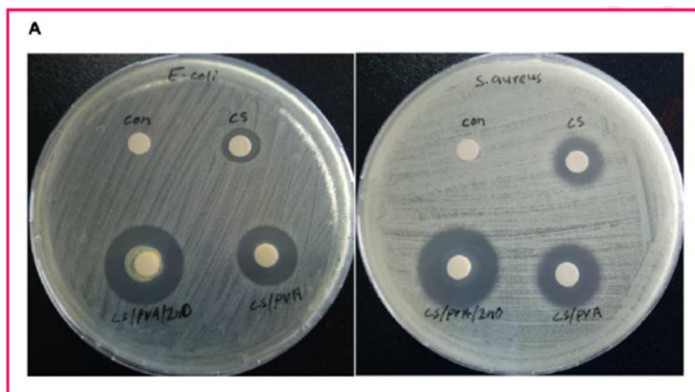

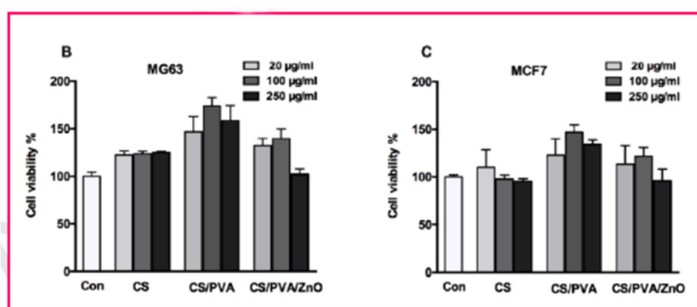

**Figure 23.** The antibacterial activities of CS, CS/PVA, and CS/PVA/ZnO: (**A**) Diameter of zone of inhibition on *E. coli* culture (gram negative bacteria); (**B**) Effect of mammalian cell viability of freshly prepared CS, CS/PVA, and CS/PVA/ZnO on MG63 (human osteosarcoma cell line) and (**C**) MCF7 (Human breast cancer cell line) cells. Reprinted/adapted with permission from Ref. [204]. Copyright© 2022, Elsevier B.V.

## 5. Future Scope and Conclusions

### 5.1. Future Scope

Although ZnO-based nanomaterials have been applied in organic pollutants removal from water and conduct antibacterial reactions in water, there is still plenty of space for improvement. The following points are the aspects that can be improved and strengthened in the application process of ZnO-based nanomaterials in the future:

(1) Exploring strategies for changing the weak toxicity of ZnO-based nanomaterials so that they can be better used in drinking water treatment, clinical medicine, virus killing, and other fields closely related to human beings.

(2) Enhancing the ability of ZnO-based nanomaterials to respond to visible light enables them to have a wider range of applications. Visible light is one of the most abundant light sources, and a better response under visible light can maximize the use of existing energy and reduce investment.

(3) Exploring the use of ZnO-based nanomaterials for photocatalytic removal of resistant bacteria, cancer cells, and other difficult-to-remove microorganisms and pathogen cells to improve the availability of the material.

(4) Exploring stronger ZnO-based nanomaterial structures and carriers to improve recyclability and improve existing problems such as high solubility and difficulty in recycling.

### 5.2. Conclusions

This manuscript is based on recent developments in antibacterial water treatment with ZnO-based nanomaterials. Due to the increasing global requirements for water environment quality and drinking water quality, especially for the prevention and control of various epidemics, new ideas and directions are provided for our study. Therefore, the

existence of bacteria and harmful microorganisms in water is introduced in detail, and various commonly used antibacterial agents and antibacterial methods are summarized. In conclusion, different morphologies of ZnO-based nanomaterials can be effectively used against various Gram-positive and Gram-negative strains by physicochemical interactions with bacterial cells. Cell membrane damage and biocidal activity are thought to be triggered by the collective action of chemical and physical interactions. Chemical interactions leading to the production of ROS and $H_2O_2$ and the release of $Zn^{2+}$ ions from ZnO solubility have been proved to be the main cause of the above activities. Subsequently, based on this theory, an in-depth study of the antibacterial mechanism was carried out. Finally, the review also summarizes the following synthetic strategies to improve the antibacterial properties of ZnO: (1) doping of alkaline earth metals to ZnO; (2) doping of transition metals to ZnO; (3) doping of noble metals to ZnO; (4) doping of rare earth metal to ZnO; and (5) loading organic antimicrobial agents.

It can be expected that the antibacterial potential of ZnO-based nanomaterials in water treatment is very promising. Studies on ZnO-based nanomaterials continuously increased in recent years, although they still have many aspects that can be improved. In the future, we can expect more perfect ZnO-based nanomaterials to be prepared to solve more antimicrobial-related problems in water treatment.

**Author Contributions:** Z.X.: Investigation, Writing—Original Draft. Q.H.: Investigation. S.W.: Investigation. X.H.: Supervision, Writing—Reviewing and Editing. Z.F.: Supervision, Writing—Reviewing and Editing. X.X.: Supervision, Writing—Reviewing and Editing. X.Z.: Supervision, Writing—Reviewing and Editing. All authors have read and agreed to the published version of the manuscript.

**Funding:** This research was supported by the National Key Research and Development Plan, China (2019YFC1907204). We are grateful for the test services from the Analytical and Testing Center of Northeastern University.

**Institutional Review Board Statement:** Not applicable.

**Informed Consent Statement:** Not applicable.

**Data Availability Statement:** Not applicable.

**Conflicts of Interest:** The authors declare no conflict of interest.

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
