# Peer review of "Recent Progress in ZnO-Based Nanostructures for Photocatalytic Antimicrobial in Water Treatment: A Review"

_applsci, doi:10.3390/app12157910_

Round 1

Reviewer 1 Report

After reading this review, I have gained a lot of information about this field.

Since it is a review, I wonder if the authors got the permission to report the figures in their manuscript.

I have few comments:

Page 6. Line 156, in Figure 4. The meaning of some symbols is not described, for example 5Z, 7Na, 8Na, 7K, 8K, etc …..

Page 7, line 177, caption of Figure 5 is not completed , the authors reported values of concentrations of what?

Page 16, line 436, please delete, means Thence will be hence

For the references list, the authors have used et al. the reviewer did not know why they did not mention the full names of the authors.

Author Response

The manuscript has be modified as your comments as follows:

  1. Page 6. Line 156, in Figure 4. The meaning of some symbols is not described, for example 5Z, 7Na, 8Na, 7K, 8K, etc …..

We have added a detailed explanation of the symbols in Figure 4 in Page 6 Line 178.

  1. Page 7, line 177, caption of Figure 5 is not completed, the authors reported values of concentrations of what?

We've completed the caption of Figure 5 and added an explanation of value in Page 7 Line 205.

  1. Page 16, line 436, please delete, means Thence will be hence

We're sorry for the spelling problems in writing the manuscript, we've corrected Thence to Hence in Page 18 Line 505.

  1. For the references list, the authors have used et al. the reviewer did not know why they did not mention the full names of the authors.

Thank you very much for your advice. We have revised the reference format and unified all reference formats into the content of the journal requirements.

Reviewer 2 Report

This is an interesting manuscript about the bactericidal activity of ZnO through the photocatalysis but some aspects should be revised before to acceptance.

The authors refer different methods to prepare the ZnO nanostructures but not explore in which way the preparation parameters of each method can affect the bactericidal activity. This should be discussed.

Table 2 should be revised since the presence of bacteria is not only the unique species of study.

Moreover, in accordance to the title of the work the authors should only discuss for bacteria. Or at least change the title to other pathogens.

In article should be clearer why to only focus on ZnO, which advantages can be found when compared with other semiconductors.

In terms of radiation source analysis, a section should be added it, since this can affect activity of ZnO, moreover different studies can be found in the literature with different radiation sources.

Author Response

We have revised the manuscript as your suggestions:

  1. The authors refer different methods to prepare the ZnO nanostructures but not explore in which way the preparation parameters of each method can affect the bactericidal activity. This should be discussed.

We are grateful for the suggestion. We have added some arguments in Section 2.1-2.4 on "how the preparation parameters of each method affect bactericidal activity". (Page 4 Line 112-120; Page 5 Line 136-143; Page 6 Line165-173; Page 7 Line 199-203)

  1. Table 2 should be revised since the presence of bacteria is not only the unique species of study.

Thank you for your suggestion. We have made minor adjustments to the subjects of our manuscript based on your comments (adjusting "antibacterial" to "antimicrobial"), which makes the subjects in Table 2 more relevant to the manuscript.

  1. Moreover, in accordance to the title of the work the authors should only discuss for bacteria. Or at least change the title to other pathogens.

Appreciate your comments. We've changed "antibacterial" to "antimicrobial" in the title. Simultaneously, we adjusted the research object in the manuscript to "antimicrobial" and discussed bacteria as part of the overall research object.

  1. In article should be clearer why to only focus on ZnO, which advantages can be found when compared with other semiconductors.

We are grateful for the suggestion. We have added some discussion in line 234-241 about the advantages of ZnO as an antimicrobial agent compared to other semiconducting materials, and based on these discussions, we have explained more clearly why ZnO was chosen as a research object.

  1. In terms of radiation source analysis, a section should be added it, since this can affect activity of ZnO, moreover different studies can be found in the literature with different radiation sources.

We appreciate the meaningful suggestion. We have added Section 4.3 (line 439-470) to explore the effects of radiation on the photocatalytic and antimicrobial properties of ZnO-based nanomaterials.
